# Characteristics of four natural poliovirus type 1 variants with six-nucleotide deletion (2,783–2,788 nt) in the VP1 region

Jie Lin,[1] Lei Zhou,[1] Chenglin Zhu,[2] Jinhang Wei,[3] Bo Lv,[4] Yuan Si,[5] Shuangli Zhu,[1] Tianjiao Ji,[1] Dongyan Wang,[1] Qian Yang,[1] Jinbo Xiao,[1] Lan Yang,[1] Kaitao Xiao,[2] Kexin Shao,[2] Yong Zhang,[1] Dongmei Yan[1]

**ABSTRACT**   Since the establishment of the Chinese acute flaccid paralysis (AFP) case surveillance system in 1999, around 7,200 strains of poliovirus (PV) type 1 have been identified and isolated. Among these, the VP1 region of 5,649 strains has been sequenced. Based on the existing VP1 region sequence library, four strains of type 1 PV with six-nucleotide deletion in the VP1 region, identified from AFP cases, healthy children, and environmental sewage samples, were identified, and their biological characteristics were investigated. Whole-genome sequence analysis showed that the similarity with the Sabin 1 strain was 99.5–99.8%, and the mutation rate in the VP1 region was only 0.11–0.55%, indicating that these strains are not vaccine-derived PVs. The missing nucleotide is located at positions 2,783–2,788 in the VP1 region, resulting in the deletion of amino acids 102 and 103 at neutralizing antigen site 1 (N-Ag I) in the BC loop. It is worth noting that the neutralization test results showed that the two strains detected from AFP and healthy children evaded immune recognition, whereas the other two from environmental sewage did not. Molecular docking and neutralization antigen site analyzes indicate that the deletion of nucleotides 2,783–2,788 in N-Ag I is not a critical factor leading to the development of neutralization escape variants.

**IMPORTANCE**   Interestingly, we observed that the VP3-60 mutation in N-Ag IIIa may be the main reason for the immune evasion of these two viruses. In addition, based on the temperature sensitivity experiments, the four viruses exhibited similar temperature sensitivity to the Sabin 1 strain and their replication ability at 39.5°C was comparable to vaccine-derived polioviruses. Although variants with six-nucleotide deletion (2,783–2,788 nt) in the VP1 region do not cause significant biological changes, they can still spread in the environment and among populations, posing a certain risk of transmission.

**KEYWORDS**    poliovirus type 1, deletion, VP1

Polioviruses (PVs) belong to the species Enterovirus C (Family Picornaviridae, Genus Enterovirus) (1). The three serotypes of PV are the etiological agents of poliomyelitis (polio), an acute infectious disease that seriously affects the health of children. The viral genome is a positive-strand RNA enclosed in a capsid composed of 60 copies of each of the four structural polypeptides: VP1, VP2, VP3, and VP4 (2). Like other RNA viruses, PV is prone to mutations during replication due to the combination of a lack of proofreading activity of the viral polymerase and the presence of lower-fidelity variants that favor the generation of deletions, which also facilitates recombination with other enteroviruses type C (3, 4), leading to the emergence of neutralization-escape variants.

Nucleotide deletions are commonly observed in various viruses. In enteroviruses, Tokarz et al. found three nucleotide (nt) deletions in the VP1 region of EV-D68 evolution branch A, which is believed to play a role in modulating the immune response to the virus and may contribute to its persistence (5). In addition, Coxsackievirus B3

**Peer Reviewer** Nora M. Chapman, University of Nebraska Medical Center College of Medicine, Omaha, Nebraska, USA

Address correspondence to Dongmei Yan, dongmeiyan1976@163.com, or Yong Zhang, yongzhang75@sina.com.

The authors declare no conflict of interest.

(CVB3) exhibits natural deletions at the 5′ end during replication in mouse hearts, with a deletion range of 7–49 nt. These novel natural CVB3 strains with 5′ end deletions (TD) are referred to as CVB3TD, and although they have replication ability, their replication rate is very slow (6). Nucleotide deletions are also found in non-enteroviruses, such as defective-interfering (DI) particles of the influenza virus, which seem to mediate protection by enhancing humoral immune responses in infected mice, rather than through self-interference with wild-type virus replication (7). In the measles virus, defective-interfering RNA (DI-RNA) can form during negative-strand RNA virus infections and amplify the full-length viral genome, thereby modulating the severity and duration of infection (8). During acute HIV-1 infection, defective proviruses rapidly accumulate, posing a major barrier to curing HIV-1 infection in individuals (9).

Unlike many RNA viruses, PV is considered a relatively stable virus with minimal changes in neutralizing epitopes and resultant amino acid substitutions (10). Continuous deletion of amino acids in the coding regions of structural proteins is extremely rare, with some exceptions. In 1991, Couderc et al. (11) isolated a neutralization escape mutant with a deletion of the entire BC loop, including N-Ag I, between positions 93 and 104. However, this deletion variant was obtained in a laboratory experiment with an engineered virus rather than through natural mutations.

In this study, we identified four natural deletion variants of PV type 1 (PV-1 deletion variants) that originated from the Sabin 1 strain and do not belong to vaccine-derived PVs (VDPVs). These strains were isolated from acute flaccid paralysis (AFP) cases, healthy children, and environmental sewage. They exhibited a hexanucleotide (2,783–2,788 nt) deletion in the VP1 region, located in or close to N-Ag I, which gave rise to two (102 and 103) amino acid deletions in the BC loop. Two of the four natural deletion variants exhibited immune evasion. Moreover, at 39.5°C, their replication capacity was comparable to that of VDPVs. This finding suggests that deletion variants present a transmission risk within populations and the environment, highlighting the need for continuous surveillance of PVs in both populations and the environment.

## RESULTS

### Sample collection

One PV-1 deletion variant was isolated from one AFP case (CHN-15057) in Shaanxi in 2011, two from two sewage samples (CHN-E10-202 and CHN-E13-180) in Heilongjiang in 2011 and 2013, and one from a healthy child (CHN-28009C) in Guangxi in 2024 (Table 1). VP1 sequence alignments showed that the strains had one, three, five, and five nucleotide substitutions in the VP1 region compared with the Sabin 1 vaccine strain (GenBank accession number AY184219), and the nucleotide substitution rates were 0.11%, 0.33%, 0.55%, and 0.55%, respectively. The WHO defines VDPV1 as type 1 PV that differs from the Sabin 1 strain by >1.0% in the VP1 region. These variants did not meet this definition and could not be classified as VDPV1.

### Full-length genomic characterization and recombination

The complete genome sequences (7,335 nt) of the four PV-1 deletion variants, obtained using the Sanger method, revealed several intriguing genetic characteristics. The overall sequence identity with Sabin 1 ranged from 99.5% to 99.8%.

Most importantly, there was a hexanucleotide in-frame deletion between positions 2,783 and 2,788, resulting in a two-amino-acid deletion at positions 102 and 103 in VP1, which are located in N-Ag I.

Among the known neurovirulence determinants of type 1 PVs (12–15), in positions 480 and 525 located in the 5′ UTR, the major determinant of the attenuated phenotype, CHN-28009C, had reverted (G-A) in position 480, while CHN-E13-180 had mutated (T-C) in position 525. The others maintained their attenuated character. The four PV-1 deletion variants did not revert at positions 935, 2,438, or 2,879. At nucleotide position 2,795 in VP1, only CHN-28009C had reverted (A-G). At position 6,203 in 3C, CHN-15057 had reverted (C-T), and all PV-1 deletion variants were non-recombinants (Table 2).

**TABLE 1** Details of the four variants with six-nucleotide deletion in the VP1 region

| Virus | Time | Place | Vaccination doses | Source | No. (%) of *VP1* nucleotide substitutions | Virus type | Accession number |
|---|---|---|---|---|---|---|---|
| CHN-15057 | 2011 | Shaanxi | 3 | AFP | 1 (0.11) | Type 1 | NMDCN0007P41 |
| CHN-E10-202 | 2011 | Heilongjiang | –[a] | Environmental sewage | 3 (0.33) | Type 1 | NMDCN0007P42 |
| CHN-E13-180 | 2013 | Heilongjiang | – | Environmental sewage | 5 (0.55) | Type 1 | NMDCN0007P43 |
| CHN-28009C | 2024 | Guangxi | 3 | Healthy child | 5 (0.55) | Type 1 | NMDCN0007P44 |

[a]"–" indicates not applicable.

## Binding affinity of PV-1 deletion variants with polio antibody

Amino acid residues 102 and 103 were eliminated from the N-Ag I of the VP1 protein, leading to a denser neutralization epitope (Fig. 1). Structural simulation of the VP1 coding sequence of Sabin 1 revealed that residues 102 and 103 were located within the BC loop (Fig. 2). Although the deletion of amino acids 102–103 caused a structural change in NAg-I amino acids 95–110 from a loose to a tight conformation, this deletion did not alter the overall structure of the BC loop . Therefore, the deletion of amino acids 102–103 in the VP1 region does not significantly affect the binding of the PV-1 deletion variant with neutralizing antibodies. Additionally, the comparison between RMSD values of PV-1 deletion variants complexed with the neutralizing antibody of P1/Mahoney PV and that of VP1 of the Sabin 1-antibody complex indicates that all PV-1 deletion variants bind to the same region of the neutralizing antibody as VP1 of Sabin 1 (Table 3), indicating that the same antibody can also fully neutralize any PV-1 deletion variants. Furthermore, the binding energy values exhibit more or less similar binding strength for VP1 of all PV-1 deletion variants with the neutralizing antibody (Fig. 3). Importantly, the deletion of amino acids 102–103 did not facilitate the emergence of neutralization-escape variants.

## Changes in neutralizing antigenic sites

The amino acid sequences within or near the predicted neutralizing antigenic (N-Ag) sites (17, 18) of the Sabin 1 strain, its parental Mahoney strain, EN-9730515 strain, and representative CHN-8184 type 1 cVDPVs were aligned with the four PV-1 deletion variants (Fig. 4). The four PV-1 deletion variants exhibited a deletion of two amino acids at N-Ag I: residues 102–103, located in the BC loop (11). In addition, strains CHN-15057 and CHN-E13-180 showed another amino acid replacement at N-AgI (VP1–90: Ile-to-Met), whereas strains CHN-E10-202 and CHN-28900C were substituted at the same position (VP1–90: Ile-to-Leu). Furthermore, at N-Ag III a, strains CHN-E10-202 and CHN-E13-180 exhibited the same characteristics as Sabin 1 at VP3–60. These amino acid

**TABLE 2** Genetic and phenotypic characterization of PV-1 deletion variants[a,c]

| Virus | Region | | | | | | | | Recombination with EV-C |
|---|---|---|---|---|---|---|---|---|---|
| | 5′UTR | | VP4 | VP3 | VP1 | | | 3D | |
| | 480 nt | 525 nt | 935 nt | 2,438 nt | 2,783–2,788 nt | 2,795 nt | 2,879 nt | 6,203 nt | |
| Sabin 1 | G | T | T | A | GATAAG | A | T | C | –[b] |
| CHN-15057 | * | * | * | * | Deletion | * | * | T | No |
| CHN-E10-202 | * | * | * | * | Deletion | * | * | * | No |
| CHN-E13-180 | * | C | * | * | Deletion | * | * | * | No |
| CHN-28009C | A | * | * | * | Deletion | G | * | * | No |
| EN-9730515 | * | * | * | * | Deletion | * | * | Unknown | Unknown |
| CHN-8184 | * | * | * | T | * | G | * | * | No |

[a]A measles patient from Turkey in 1997 (EN-9730515, GenBank accession number: AF065158) (16). The first type 1 Circulating Vaccine-Derived Polio Virus (cVDPVs) strain identified in China in 2004 (CHN-8184 Gen bank accession number FJ769378).
[b]"–" indicates not applicable.
[c]* indicates the same nucleotide as in Sabin 1.

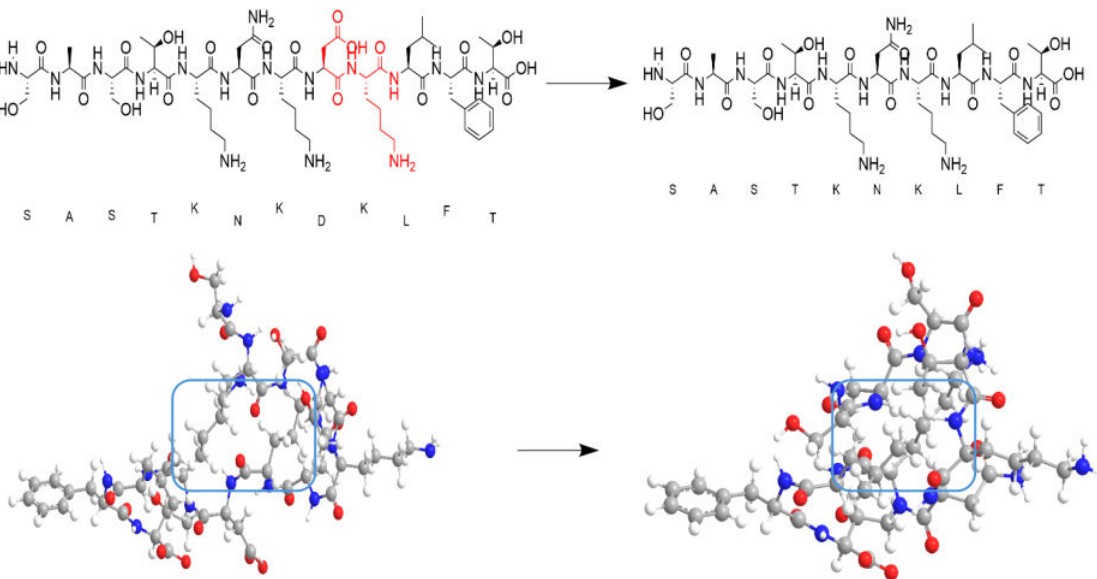

**FIG 1** Changes in the molecular structure of N-Ag I deletion 102–103 amino acids.

replacements in the epitopes, especially at N-Ag I and N-Ag IIIa, may be responsible for the aberrant results of the neutralizing test (Table 4).

## Neutralization assays with human sera

100% of the sera could neutralize the four PV-1 deletion variants. In the serum of 3-month-old children (vaccinated with inactivated polio vaccine [IPV]), the geometric mean titers (GMTs) for Sabin 1, CHN-15057, CHN-E10-202, CHN-E13-180, CHN-28009C,

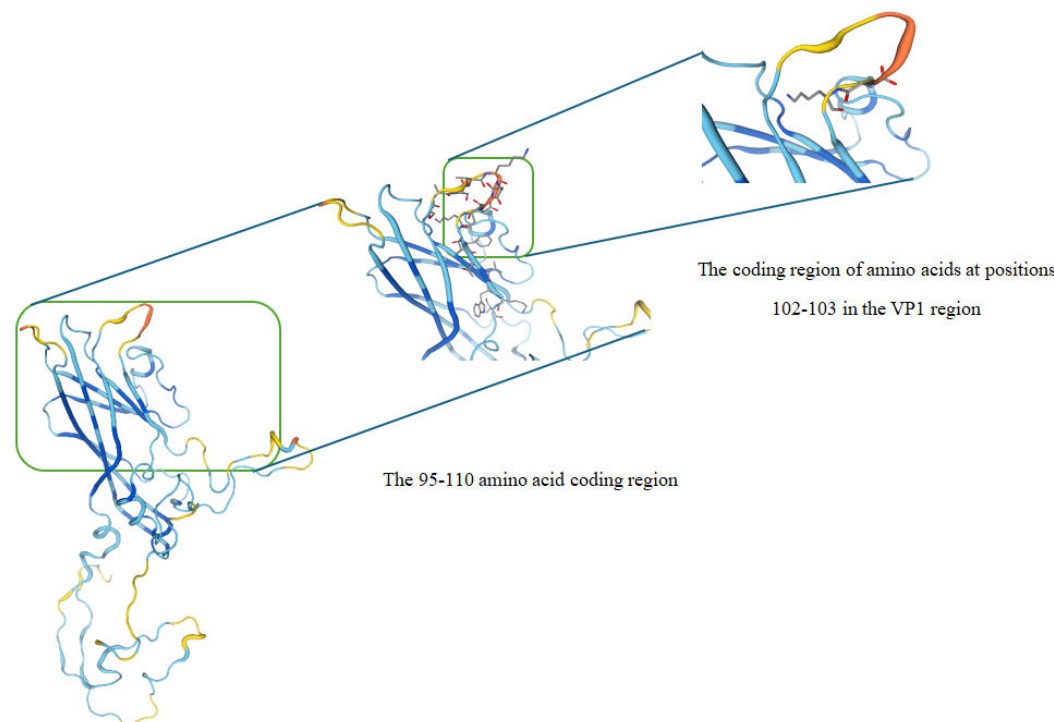

The coding region of amino acids at positions 102-103 in the VP1 region

The 95-110 amino acid coding region

Structure of the protein encoded by the VP1 region of the Sabin 1 strain

**FIG 2** The VP1 of Sabin 1 encodes a structural protein.

**TABLE 3** Comparison of RMSD (root mean square deviation) and viral antigen-antibody binding energy values of PV-1 deletion variants and Sabin 1

| Model/ Ag-Ab complex | Ag/Ab binding energy (kcal/mol) | RMSD in Angstrom against PVS/Ab complex |
|---|---|---|
| Sabin1 | −632 | 0 |
| CHN-15057 | −630 | 1.61 |
| CHN-E10-202 | −634 | 1.63 |
| CHN-E13-180 | −631 | 1.63 |
| CHN-28009C | −631 | 1.62 |

and CHN-8184 were 904.7, 107.7, 1,198.6, 1,327.9, 48.2, and 68.3, respectively. In the serum of children aged 0–6 years from Leibo County, the GMTs were 455.4, 177.3, 369.3, 394.1, 105.9, and 155.2 (Table 4). GMTs of the deletion variants were compared using the Wilcoxon signed-rank test. The results demonstrated that the GMTs of CHN-15057, CHN-28009C, and CHN-8184 were significantly lower than that of Sabin 1 ($P < 0.05$). In contrast, the GMTs of CHN-E10-202 and CHN-E13-180 were comparable to that of Sabin 1, with no statistically significant difference ($P > 0.05$). This suggests that deletion variants from AFP and healthy children escaped immunity, while deletion variants isolated from environmental sewage did not. This phenomenon can be attributed to mutations in N-Ag IIIa (VP3–60) rather than the deletion of 102–103 amino acids at N-Ag I (Fig. 4), resulting from repeated replication cycles of deleted variants within the human body.

## Temperature sensitivity of the PV-1 deletion variants

Temperature sensitivity usually serves as an *in vitro* marker for the attenuation of PV vaccine strains; although the link between temperature sensitivity and attenuation may not be straightforward, it could serve as an indicator of virulence in enteroviruses (19). To investigate viral growth characteristics, we examined the replication of this virus at two different temperatures and constructed a one-step growth curve (Fig. 5). The titers of all strains cultured at 36.5°C and 39.5°C indicated temperature sensitivity of the strain, as the titer reduction exceeded two logarithms at these temperatures. However, at 39°C, the replication ability of PV-1 deletion variants closely resembling the cVDPV CHN-8184 strain. The titers of the two other strains, CHN-E10-202 and CHN-E13-180, decreased after 48 h. At high temperatures, the deletion of 102–103 amino acids might affect the replication ability of the virus.

## DISCUSSION

The era of wild PV is rapidly ending. In the short term, the only source of global PV infection appears to be the OPV, making it particularly important to distinguish between vaccinated and non-vaccinated strains. Large-scale polio outbreaks often occur because of the high proportion of antigenic epitopes. For example, an outbreak of an imported type 1 wild PV from Pakistan occurred in the Xinjiang Uygur Autonomous

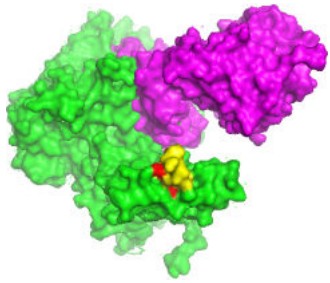

**FIG 3** Simulated PV-1 deletion variants demonstrate binding to neutralizing antibody for VP1. Green represents the PV-1 deletion variants, purple denotes the neutralizing antibody for VP1, yellow indicates N-Ag I, and red highlights the amino acid deletion positions at 102–103.

**FIG 4** Alignment of amino acid residues of neutralizing antigenic (N-AG) sites. N-Ag site 1 (VP1: 88–106), site 2 (VP2: 164–173; VP2: 269–271; VP1: 221–226), site 3a (VP3: 54–61; VP3: 70–74; VP1: 287–292), and site 3b (VP2: 71–73; VP3: 75–79) for Sabin 1. PV-1 deletion variants and the CHN-8184 strain. The EN-9730515 Sabin-Like strain was cited from a previous report.

Region of China in 2011, and genomic sequencing indicated that the Xinjiang strain was greatly distinguishable from the Sabin 1 strain in its neutralizing antigenic sites (20). Furthermore, Hughes et al. (21) reported an unprecedented evolution of PV type 3 during an outbreak in Finland from August 1984 to January 1985, resulting in a high proportion of mutations in neutralizing epitopes without deletions and giving rise to a neutralization-escape mutant. Similarly, Drexler et al. (22) reported an outbreak of type 1 PV in the Democratic Republic of the Congo from 2010 to 2011, caused by a PV strain with significant mutations in antigenic sites. In this study, we identified four PV variants with six-nucleotide deletion (2,783–2,788 nt) in the VP1 region with a deletion of six nucleotides located at or near N-Ag I, leading to the loss of two amino acids (102–103) in the BC loop. These samples were isolated from healthy children, AFP cases, and environmental sewage, indicating the widespread presence of this deleted PV variant in nature, with a certain capability for transmission and replication, allowing its spread between the environment and populations.

The four PV-1 deletion variants were derived from the Sabin 1 strain and did not belong to VDPVs, as seen in the whole-genome sequence analysis. Among the key attenuation sites, only the CHN-28009C strain was restored to G-A at position 480, whereas the CHN-E13-180 strain was mutated to T-C at position 525. In addition, we cannot speculate whether the hexanucleotide deletion in N-Ag I had any effect on neurovirulence, because the neurovirulence of these deletion variants has not yet been tested.

Moreover, molecular docking analysis and neutralizing antigen site analysis showed that the deletion of nucleotides 2,783–2,788 in N-Ag I was not the key to the emergence of neutralization-escape variants. However, neutralization tests revealed that both CHN-15057 and CHN-28009C evolved into immune escape variants. Notably, the

**TABLE 4** Neutralization assays with human sera[a]

| | Serum source | | | |
|---|---|---|---|---|
| | Three-month-old children's serum (IPV) | | 0- to 6-old children's serum from Leibo County | |
| Virus | Seropositive subjects, No. (%) | GMT | Seropositive subjects, No. (%) | GMT |
| Sabin1 | 100 | 904.7 | 100 | 455.4 |
| CHN-15057 | 100 | 107.7 | 100 | 177.3 |
| CHN-E10-202 | 100 | 1,198.6 | 100 | 369.3 |
| CHN-E13-180 | 100 | 1,327.9 | 100 | 394.1 |
| CHN-28009C | 100 | 48.2 | 100 | 105.9 |
| CHN-8184 | 100 | 68.3 | 100 | 155.2 |

[a]Sera were obtained from individuals vaccinated with oral polio vaccine (OPV) or IPV, and their neutralization capacity against PV-1 deletion variants was compared to that against Sabin one and CHN-8184.

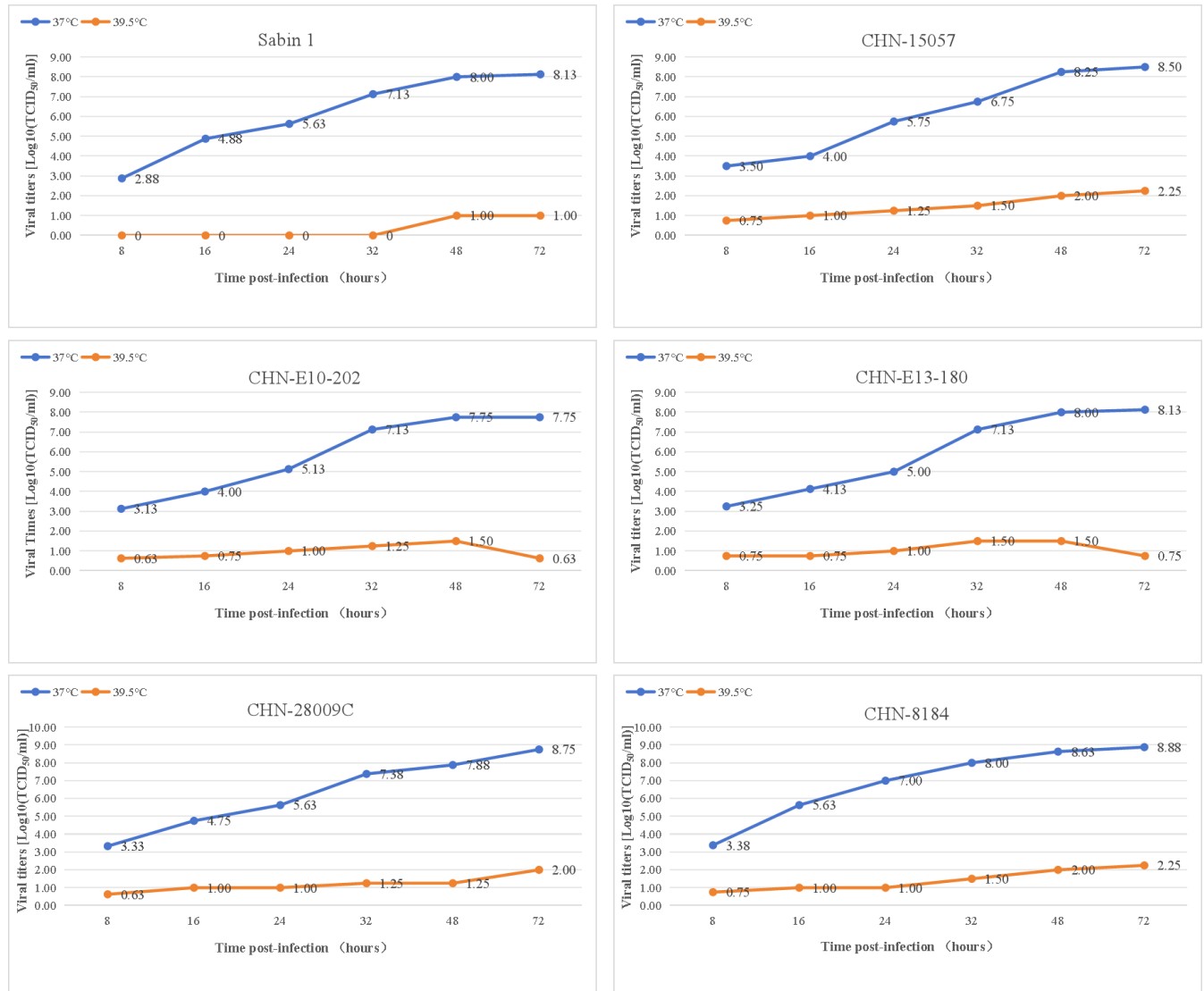

**FIG 5** Temperature sensitivity test curves.

CHN-E10-202 and CHN-E13-180 strains isolated from environmental sewage were still effectively protected by vaccination. This phenomenon may be attributed to mutations in N-Ag IIIa (VP3–60) resulting from repeated replication cycles of the deletion mutant within the human body, leading to variations in neutralization efficacy. According to the report by Mulders et al. (16) on PV type 1 isolated from a measles patient (16), we found that amino acids 102–103 in the VP1 region of strain 97-30515 exhibited the same deletion. In the micro-neutralization test, strain 97-30515 was not neutralized by monoclonal antibodies 956 (specific to site 1), 423, 3D8, 1D4E8, or 1D7 (all specific to site 3). Additionally, in the neutralization test with serum, the results showed that the neutralization titer was similar to that of the prototype vaccine strains Sabin 1 and Mahoney, which differed from strains CHN-15057 and CHN-28009C in this study. Furthermore, the deleted 102–103 amino acid residues are also located at the binding site of the C3 neutralizing antibody (23, 24), which may lead to the inability of the C3 neutralizing antibody to effectively neutralize the virus, thereby facilitating viral invasion.

The most significant feature of this virus is its main capsid protein, VP1. Currently, we do not yet understand the specific formation process of hexanucleotide deletions in N-Ag I; however, we speculate that there may be two reasons for this. First, it may be due

to imprecise homologous recombination triggering genome rearrangement; during the RNA polymerase chain-switching process, a hexanucleotide leap can occur (3). However, it is worth noting that recombination analysis did not find recombination between these deletion variants and type C enterovirus. Second, PV DI particles were originally produced by type 1 PV, which lacks 15–20% of the viral genome. This phenomenon is due to the production of the wild-type PV after multiple passages under high multiple of infection conditions (25, 26). Similarly, vaccine strains can also produce DI but require more passages (27). Therefore, we speculate that these deletion variants may be precursors of type 1 deletion particles. Their production may be the result of several replication cycles within an individual, due to repeated passages in the population, or the result of viral replication in individuals with low levels of immunity.

From the perspective of viral replication, the missing amino acids at positions 102–103 are aspartic acid and lysine, both of which are hydrophilic amino acids. After the deletion, the structure of the NAg I molecule becomes tighter. Hydrophilic amino acids are usually located on the surface of proteins (28), and their loss may lead to a decrease in the exposure level of antigens and potentially affect the binding of the virus to cell receptors, thereby affecting viral replication. However, growth curves from different temperatures show that deletion variants remained temperature sensitive. Moreover, the replication ability did not change significantly at 37°C, but was closely resembling the cVDPVs CHN-8184 strain at 39°C, which is similar to the results of Kaplan and Racaniello (29) This study suggests that even if the entire P1 region is deleted, the replication ability of PV will still not be affected.

The discovery of these PV variants with six-nucleotide deletion (2,783–2,788 nt) in the VP1 region indicates that the deletion of amino acids 102–103 in N-Ag I of PV type 1 does not result in immune escape. We propose that this deletion is not a critical factor for immune evasion; rather, the mutation at VP3-60 in N-Ag IIIa may serve as the primary driver of immune escape. However, these deletion variants can exist in the environment and among individuals, and their replication ability is equivalent to that of VDPVs, which may pose a certain risk of transmission. Furthermore, the potential impact of this deletion on neurotoxicity remains uncertain as we did not conduct experiments to evaluate its neurotoxic effects. Therefore, the emergence of deletion variants may have an impact on the global polio eradication plan, warning us of the importance of long-term monitoring and vaccination of the population and environment to achieve the goal of ending polio and eventually eliminating the disease.

## MATERIALS AND METHODS

### Sample collection

Since the establishment of the AFP surveillance network in China in 1999, approximately 7,200 strains of PV type 1 have been identified and isolated, with the VP1 region sequenced from 5,649 of these strains. We screened these strains in the VP1 database and found that four had nucleotide deletions in the VP1 region.

To assess whether the deletion variants can be effectively protected against by vaccines, we planned to conduct neutralization experiments using a single dose of an IPV or OPV (30). However, since 2016, China has revised its poliovaccination policy, making it impossible to collect serum samples solely from individuals vaccinated with OPV. Therefore, in the serum neutralization experiment, 40 serum samples were collected from healthy children, including 20 serum samples from 3-month-old children who were administered a single dose of IPV. The remaining 20 samples were obtained from healthy children aged 0–6 years in Leibo County, Sichuan Province, China, where an outbreak of type 2 circulating vaccine-derived PV occurred between April 2018 and May 2019 (31). We obtained informed consent from all children's parents during the serum collection process; none of the children showed any disease characteristics at the time of serum collection.

Furthermore, no wild strains of PV type 1 are present in China. The biological characteristics of type 1 cVDPVs, which first emerged in Aba Prefecture, Sichuan Province, in 2004, were similar to those reported by Mahoney (32). Consequently, CHN-8184 was selected as the reference strain for both neutralization and temperature-sensitive experiments. Data are deposited in the National Microbiology Data Center (NMDC) (https://nmdc.cn/resource/genomics/sequence) with accession numbers NMDCN0007P41, NMDCN0007P42, NMDCN0007P43, and NMDCN000744.

## Full-length genome sequencing and recombination analysis

Viral RNA was extracted from the PV isolates using the QIAamp Mini Viral RNA Extraction Kit (Qiagen) and used for RT-PCR amplification using standard methods (33, 34). The reaction system was prepared using the One Step PrimeScript RT-PCR Kit (TaKaRa), and the primer sequences of the whole genome were obtained from Harrington et al. (35–37). The PCR products were purified using a QIAquick Gel Extraction Kit (Qiagen) and sequenced using the Sanger method.

BLAST server was used to analyze the 5′ UTR, P2, and P3 coding region sequences of strains and to compare them with the sequences in GenBank. Sequences with more than 85% similarity to the 5′ UTR, P2, and P3 regions of strains, respectively, were identified as potential parents of the strain and were downloaded from GenBank. A Recombinant Detection Program (RDP, version 4.46) was used to screen for recombination signals (38).

## Molecular docking analysis for antigen-antibody affinity

Amino acid sequences of VP1 and VP1 antigen site 1 from Sabin 1 and PV-1 deletion variants were used to build a three-dimensional protein model using the SWISS MODEL server and Chem Draw 20.0, a web-based integrated service for protein structure homology modeling and molecular simulation (39). All VP1 sequences shared >98% sequence identity with the template protein of P1/Mahoney PV (Protein Data Bank [PDB] code 1AR7), with 95% query coverage. The global quality of the generated models was assessed using the QMEAN scoring function. After structure validation of modeled VP1, molecular docking studies were performed using the PyMOL (a tool used for protein docking), and the corresponding Ag/Ab Binding Energy and RMSD value were calculated. Each validated structure of VP1 was separately docked with neutralizing antibody for VP1 (PDB code 1FPT), and the docking modes with the lowest binding energy were considered for interaction analysis (40).

## Neutralizing test

After viral culture purification, the viral titer was measured three times and the average value was obtained, followed by the determination of serum antibody titers using a neutralization test (41). In total, 40 serum samples were inactivated at 56°C for 30 min, and then diluted with 10 gradients from 1:4 to 1:1024 (1:4, 1:8, 1:16, 1:32, 1:64, 1:128, 1:256, 1:512, 1:1,024, and 1:2,048) for detection. Next, 50 µL of each serum dilution was added to a 96-well plate and 50 µL of virus was added to each well, which confirmed that each well contained 100 TCID50. The 96-well plate was then placed in a 5% $CO_2$ incubator at 36.5°C for 2 h. Two wells in each column were used as serum controls. We also used a new plate for cell and virus control, which we called a virus back-titration. Next, 100 µL of RD cells was added to each well and incubated at 36.5°C with 5% $CO_2$ for 5 days, with intermittent observation (42).

## Temperature sensitivity

The temperature sensitivity phenotype of the viruses was evaluated by titrating the same virus stocks at 36.5°C and 39.5°C, with samples taken at five time points after infection (8, 16, 24, 32, 48, and 72 h). Virus titers were evaluated in RD cells by determining the number of 50% tissue culture infective dose units ($TCID_{50}$) per mL, following the WHO

standard protocol. Viral isolates that showed a drop of more than 2 log at different temperatures were considered temperature-sensitive (43).

## ACKNOWLEDGMENTS

This work was supported by the National Key Research and Development Program of China [grant number 2024YFC2310400]. We thank all members of the Zhang lab for their support.

## AUTHOR AFFILIATIONS

[1]National Polio Laboratory, WHO WPRO Regional Polio Reference Laboratory, National Health Commission Key Laboratory for Biosecurity, National Health Commission Key Laboratory for Medical Virology, National Institute for Viral Disease Control and Prevention, Chinese Center for Disease Control and Prevention, Beijing, China
[2]School of Public Health, Shandong First Medical University & Shandong Academy of Medical Sciences, Jinan, China
[3]Guangxi Center for Disease Control and Prevention, Nanning, China
[4]Heilongjiang Center for Disease Control and Prevention, Harbin, China
[5]Shaanxi Center for Disease Control and Prevention, Xian, China

## AUTHOR ORCIDs

Jie Lin http://orcid.org/0009-0007-8054-3775
Yong Zhang http://orcid.org/0000-0002-2692-5437
Dongmei Yan http://orcid.org/0000-0002-2817-9431

## AUTHOR CONTRIBUTIONS

Jie Lin, Formal analysis, Validation, Writing – original draft | Lei Zhou, Formal analysis, Methodology | Chenglin Zhu, Conceptualization, Validation | Jinhang Wei, Data curation | Bo Lv, Data curation | Yuan Si, Data curation, Methodology | Shuangli Zhu, Data curation, Formal analysis | Tianjiao Ji, Investigation, Methodology | Dongyan Wang, Data curation, Methodology | Qian Yang, Resources, Software | Jinbo Xiao, Data curation | Lan Yang, Data curation, Visualization | Kaitao Xiao, Data curation, Software | Kexin Shao, Conceptualization, Data curation | Yong Zhang, Project administration, Writing – review and editing | Dongmei Yan, Funding acquisition, Writing – review and editing

## ETHICS APPROVAL

This study did not involve human participants or human experimentation; the only human materials used were stool samples collected from AFP patients and healthy children at the instigation of the Ministry of Health People's Republic of China for public health purposes, and written informed consent for the use of their clinical samples was obtained from the parents of the child patients on their behalf. This study was approved by the second session of the Ethics Review Committee of the National Institute for Viral Disease Control and Prevention, Chinese Center for Disease Control and Prevention, and the methods were performed in accordance with the approved guidelines.

## ADDITIONAL FILES

The following material is available online.

Open Peer Review

**PEER REVIEW HISTORY (review-history.pdf).** An accounting of the reviewer comments and feedback.

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
