## [Reviewer comments · Microbiology Spectrum]

Microbiology Spectrum

Characteristics of four natural poliovirus type 1 variants with six nucleotide deletion (nt 2783-2788) in the VP1 region

Jie Lin, Lei Zhou, Chenglin Zhu, Jinhang Wei, Bo Lv, Yuan Si, Shuangli Zhu, Tianjiao Ji, Dong Wang, Qian Yang, Jinbo Xiao, Lan Yang, Kaitao Xiao, Kexin Shao, Dongmei Yan, and Yong Zhang

Corresponding Author(s): Dongmei Yan, Chinese Center for Disease Control and Prevention

Review Timeline:

Submission Date:	April 28, 2025
Editorial Decision:	June 24, 2025
Revision Received:	July 7, 2025
Accepted:	July 9, 2025

Editor: Takamasa Ueno

Reviewer(s): Disclosure of reviewer identity is with reference to reviewer comments included in decision letter(s). The following individuals involved in review of your submission have agreed to reveal their identity: Nora M. Chapman (Reviewer #1)

Transaction Report:

DOI: <https://doi.org/10.1128/spectrum.01334-25>

Re: Spectrum01334-25 (**Characteristics of four natural poliovirus type 1 variants with six nucleotide deletion (nt 2783-2788) in the VP1 region**)

Dear Ms. Dongmei Yan:

Thank you for the privilege of reviewing your work. Below you will find my comments, instructions from the Spectrum editorial office, and the reviewer comments.

Revision Guidelines

Sincerely,
Takamasa Ueno
Editor
Microbiology Spectrum

Reviewer #1 (Comments for the Author):

This is a report of detection of poliovirus type 1 (PV1) isolates with a deletion (aa 1101,1102) in the neutralizing antigen site 1 (NAg-I), a rare variant. This has been reported before as the authors note (reference 15 in Table 2; in an encephalitis patient) and in a report on persisting PV1 in an immunodeficient individual in which this deletion was found (Odoom JK, Yunus Z, Dunn G, Minor PD, Martín J. Changes in population dynamics during long-term evolution of Sabin type 1 poliovirus in an

immunodeficient patient. *J Virol.* 2008 Sep;82(18):9179-90. doi: 10.1128/JVI.00468-08). I note that in reference 15, analysis of neutralization by a monoclonal antibody specific for NAg-I, found a lack of neutralization in comparison to the vaccination strain, Sabin-1. This earlier report should have been discussed in the text and in the discussion (see below). However, the rarity of this variation is demonstrated in the present study as the sequence determination of the VP1 of 5649 PV1 strains from this study of over 5,000 strains found only 4 isolates with this variation [one case of acute flaccid paralysis (AFP), two from sewage samples and one from a healthy child] and all derived from Sabin 1.

The authors found reversion of one or two of the Sabin-1 attenuation sites in 3 of the four deleted strains. And, in examination of the Sabin-1 NAg-1, all four of the deleted strains had variation in addition to the deletion of aa 1102 and 1103 and two also had variation at NAg-IIIa. The authors plotted the deletions upon the structure of the C3 monoclonal antibody bound to PV1 VP1 and state on lines 134-138: "Additionally, all PV-1 deletion variants bound to the same region of the neutralizing antibody as VP1 of Sabin 1, indicating that the same antibody can fully neutralize any PV-1 deletion variants. Importantly, the deletion of amino acids 102-103 did not facilitate the emergence of neutralization-escape variants (Fig. 3)." It should be noted in Reference 15, use of a micro-neutralizing monoclonal antibody specific for NAg-I did indicate a loss of neutralization relative to Sabin 1. Is the neutralizing antibody C23 (Reference 23) which has been shown to recognize VP1 aa 95-110? What is the basis of arguing that the neutralizing antibody can dock with the deletion mutants? This is not addressed in the Methods section, lines 306-317. The authors also presented data indicating that there was a change in structure of the BC loop of VP1 with the deletion of aa 1102 and 1103 (Figure 1 and 2). This should be indicative of a change of binding of neutralizing antibodies specific for NAg-I. If the authors disagree with this, they should make this argument clear. In both this study and in reference 15, human sera from vaccinees did neutralize the deletion variant strains but in this study (unlike reference 15), the global mean titers for the strains from a case of AFP and a healthy child were less efficiently neutralized than Sabin 1 which the authors attribute to mutations in these strains at NAg-IIIa.

The authors examined the temperature sensitivity of these strains which indicated somewhat more replication at 39.5oC but did not provide the statistical significance of the variation.

Minor issues:

Lines 47-48, please provide a reference.

Lines 119-122: "The four PV-1 deletion variants did not revert to positions 935, 2438, or 2879. At nucleotide position 2795 in VP1, only CHN-28009C had reverted (A-G). Position 6203 in 3C, CHN-15057 had reverted (C-T), and all PV-1 deletion variants were non-" should be "The four PV-1 deletion variants did not revert at positions 935, 2438, or 2879. At nucleotide position 2795 in VP1, only CHN-28009C had reverted (A-G). At position 6203 in 3C, CHN-15057 had reverted (C-T), and all PV-1 deletion variants were non-".

Lines 149-150: These amino acid replacements in the epitopes, especially at N-Ag I and 149 N-Ag III a, may be responsible for the aberrant results of the neutralizing test (Fig. 4). Fig. 4 should be Table 3.

Line 152: "The overall seroprevalence of the four PV-1 deletion variants was 100%." should be "100% of the sera could neutralize the four PV-1 deletion variants."

Lines 275-276 should be referenced. I suggest: WHO position paper - June 2022[J] *Wkly Epidemiol Rec.* 2022;25(97):277-300. In the methods section, lines 285-292: Please indicate which accession number is assigned to which of the strains. The addition of a column with the accession number to Table 1 would make this clear. I assume NMDCN0007 should be NMDCN00070P44. Reference 2 and 24 are identical and not complete: the reference should be Vadim I. Agol, *Recombination and Other Genomic Rearrangements in Picornaviruses*, 1997 *Seminars in Virology*, 8(2):77-84

In summary, this study examined the presence of a variant PV1 derived from Sabin 1 which had two aa deleted from the VP1 BC loop, thus altering a major neutralization site. This variant was found in two previous studies of PV1 present in individuals but this examination of an extended number of isolates from the Chinese population does indicate how rare this occurrence is (4 in over 5,000 isolates). The authors provided analysis of the mutation or reversion from Sabin 1 sites and an analysis of the ability of these variants to escape vaccination immunity.

Reviewer #2 (Comments for the Author):

Some puntual doubts:

- EN-9730515 strain is a wild poliovirus or Sabin?
- Table 1, sample CHN-28009C time 2024, in the text said 2019, which is correct?

Response to Reviewer 1 Comments:

Comments for the Author:

Major issues:

- 1. In reference 15, analysis of neutralization by a monoclonal antibody specific for NAg-I, found a lack of neutralization in comparison to the vaccination strain, Sabin-1. This earlier report should have been discussed in the text and in the discussion.**

----- Response to reviewer 1: Thank you for your comments. We added a description of this early report to the discussion, as described in lines 229-236: "According to the report by Mulders et al. (1999) on poliovirus type 1 isolated from a measles patient (16,24), we found that amino acids 102-103 in the VP1 region of strain 97-30515 exhibited the same deletion. In the micro-neutralization test, strain 97-30515 was not neutralized by monoclonal antibodies 956 (specific to site 1), 423, 3D8, 1D4E8, or 1D7 (all specific to site 3). Additionally, in the neutralization test with serum, the results showed that the neutralization titer was similar to that of the prototype vaccine strains Sabin 1 and Mahoney, which differed from strains CHN-15057 and CHN-28009C in this study."

- 2. The authors plotted the deletions upon the structure of the C3 monoclonal antibody bound to PV1 VP1 and state on lines 134-138: "Additionally, all PV-1 deletion variants bound to the same region of the neutralizing antibody as VP1 of Sabin 1, indicating that the same antibody can fully neutralize any PV-1 deletion variants. Importantly, the deletion of amino acids 102-103 did not facilitate the emergence of neutralization-escape variants (Fig. 3)." It should be noted in Reference 15, use of a micro-neutralizing monoclonal antibody specific for NAg-I did indicate a loss of neutralization relative to Sabin 1. Is the neutralizing antibody C3 (Reference 23) which has been shown to recognize VP1 aa 95-110? What is the basis of arguing that the neutralizing antibody can dock with the deletion mutants? This is not addressed in the Methods section, lines 306-317.**

----- Response to reviewer 1: Thank you for your comments.

- 1) In Figure 3, we conducted simulations to the binding interaction between poliovirus type 1 serum and the Sabin1 strain, rather than with the monoclonal antibody C3. The simulation outcomes indicated that the optimal binding position of the poliovirus type 1 serum to Sabin1 did not involve amino acids 102-103 in the VP1 region. Therefore, we inferred that the deletion of amino acids 102-103 in the VP1 region was not a critical determinant for the virus's immune escape mechanism.
- 2) The neutralizing antibody C3, as described in Reference 24 in this manuscript, has demonstrated the ability to recognize the VP1 amino acid residues 95-110.

The detailed epitope mapping and the significance of this recognition in the context of viral neutralization are likely elaborated in the referenced study, providing insights into the mechanism by which C3 exerts its neutralizing effects. In addition, a report also suggests that the C3 neutralizing epitope is situated in the same region. (Wychowski C, Van der Werf S, Siffert O, Crainic R, Bruneau P, Girard M. 1983. A poliovirus type 1 neutralization epitope is located within amino acid residues 93 to 104 of viral capsid polypeptide VP1. The EMBO Journal 2:2019-2024.).

3) Neutralizing antibodies can dock with PV-1 deletion variants based on Ag/Ab Binding Energy and RMSD values, We have added a Table 3(Comparison of RMSD (root mean square deviation) and viral antigen-antibody binding energy values of of PV-1 deletion variants and Sabin 1) and also described in detail in the results, as described in lines 134-141 :“Additionally, the comparison between RMSD values of PV-1 deletion variants complexed with the neutralizing antibody of P1/Mahoney poliovirus and that of VP1 of the Sabin 1-antibody complex indicates that all PV-1 deletion variants bind to the same region of the neutralizing antibody as VP1 of Sabin 1 (Table 3), indicating that the same antibody can also fully neutralize any PV-1 deletion variants. Furthermore, the binding energy values exhibit more or less similar binding strength for VP1of all PV-1 deletion variants with the neutralizing antibody (Fig. 3).”

3. **The authors also presented data indicating that there was a change in structure of the BC loop of VP1 with the deletion of aa 102 and 103 (Figure 1 and 2). This should be indicative of a change of binding of neutralizing antibodies specific for NAg-I. If the authors disagree with this, they should make this argument clear.**

----- Response to reviewer 1: Thank you for your comments. We have added a detailed description of the changes in the specific binding of neutralizing antibodies caused by the deletion of amino acids 102-103 in the VP1 region, as described in lines 130-134: “Although the deletion of amino acids 102-103 caused a structural change in NAg-I amino acids 95-110 from a loose to a tight conformation, this deletion did not alter the overall structure of the BC loop. Therefore, the deletion of amino acids 102-103 in VP1 region does not significantly affect the binding of the PV-1 deletion variant with neutralizing antibodies in the population.”

4. **The authors examined the temperature sensitivity of these strains which indicated somewhat more replication at 39.5°C but did not provide the statistical significance of the variation.**

----- Response to reviewer 1: Thank you for your comments. We conducted a variance analysis to compare the virus titers of the PV-1 deletion mutant and Sabin1 at different time points at 39.5°C. The results indicated that $P > 0.05$, meaning the difference was not statistically significant. Therefore, we have made corrections to the elements in this manuscript, as described in lines 188-189:“However, at 39°C, the replication ability of PV-1 deletion variants closely resembling the cVDPVs

CHN-8184 strain”

Minor issues:

1. Lines 47-48, please provide a reference.

----- Response to reviewer 1: Thank you for your comments. we have provide a reference in this manuscript as follows : STANWEY G. 2005. Family picornaviridae. Virus taxonomy:757-778.

2. Lines 119-122: "The four PV-1 deletion variants did not revert to positions 935, 2438, or 2879. At nucleotide position 2795 in VP1, only CHN-28009C had reverted (A-G). Position 6203 in 3C, CHN-15057 had reverted (C-T), and all PV-1 deletion variants were non-" should be "The four PV-1 deletion variants did not revert at positions 935, 2438, or 2879. At nucleotide position 2795 in VP1, only CHN-28009C had reverted (A-G). At position 6203 in 3C, CHN-15057 had reverted (C-T), and all PV-1 deletion variants were non-".

----- Response to reviewer 1: Thank you for your comments. Modifications completed.

3. Lines 149-150: These amino acid replacements in the epitopes, especially at N-Ag I and 149 N-Ag III a, may be responsible for the aberrant results of the neutralizing test (Fig. 4). Fig. 4 should be Table 3.

----- Response to reviewer 1: Thank you for your comments. Modifications completed.

4. Line 152: "The overall seroprevalence of the four PV-1 deletion variants was 100%." should be "100% of the sera could neutralize the four PV-1 deletion variants."

----- Response to reviewer 1: Thank you for your comments. Modifications completed.

5. Lines 275-276 should be referenced. I suggest: WHO position paper - June 2022[J] Wkly Epidemiol Rec. 2022;25(97):277-300.

----- Response to reviewer 1: Thank you for your comments.“WHO position paper - June 2022[J] Wkly Epidemiol Rec. 2022;25(97):277-300.” has be referenced in this manuscript.

6. In the methods section, lines 285-292: Please indicate which accession number is assigned to which of the strains. The addition of a column with the accession number to Table 1 would make this clear. I assume NMDCN0007 should be NMDCN00070P44.

----- Response to reviewer 1: Thank you for your comments.We have added a column with the accession number to Table 1 and corrected "NMDCN0007" to "NMDCN00070P44".

- 7. Reference 2 and 24 are identical and not complete: the reference should be Vadim I. Agol, Recombination and Other Genomic Rearrangements in Picornaviruses, 1997 Seminars in Virology, 8(2):77-84**

----- Response to reviewer 1: Thank you for your comments. Modifications completed.

Response to Reviewer 2 Comments:

Comments for the Author:

- 1. EN-9730515 strain is a wild poliovirus or Sabin?**

----- Response to reviewer 1: Thank you for your comments. EN-9730515 strain is a Sabin-Like strain And we marked it in the title of Figure 4, as described in lines 381-382: The EN-9730515 Sabin-Like strain was cited from a previous report.

- 2. Table 1, sample CHN-28009C time 2024, in the text said 2019, which is correct?**

----- Response to reviewer 1: Thank you for your comments. The CHN-28009C strain was isolated in 2024, we have made corrections to the contents in this manuscript, as described in lines 94-96: two from two sewage samples (CHN-E10-202 and CHN-E13-180) in Heilongjiang in 2011 and 2013, and one from a healthy child (CHN-28009C) in Guangxi in 2024 (Table 1).

Re: Spectrum01334-25R1 (**Characteristics of four natural poliovirus type 1 variants with six nucleotide deletion (nt 2783-2788) in the VP1 region**)

Dear Dr. Dongmei Yan:

Your manuscript has been accepted, and I am forwarding it to the ASM production staff for publication. Your paper will first be checked to make sure all elements meet the technical requirements. ASM staff will contact you if anything needs to be revised before copyediting and production can begin. Otherwise, you will be notified when your proofs are ready to be viewed.

Sincerely,
Takamasa Ueno
Editor
Microbiology Spectrum